# Survival Benefit of Resection Surgery for Pancreatic Ductal Adenocarcinoma with Liver Metastases: A Propensity Score-Matched SEER Database Analysis

**DOI:** 10.3390/cancers14010057

**Published:** 2021-12-23

**Authors:** Thomas M. Pausch, Xinchun Liu, Jiaqu Cui, Jishu Wei, Yi Miao, Ulrike Heger, Pascal Probst, Stephen Heap, Thilo Hackert

**Affiliations:** 1Department of General, Visceral and Transplantation Surgery, Heidelberg University Hospital, 69120 Heidelberg, Germany; Thomas.Pausch@med.uni-heidelberg.de (T.M.P.); liu.xinchun@njmu.edu.cn (X.L.); cuijiaqu@126.com (J.C.); Ulrike.Heger@med.uni-heidelberg.de (U.H.); Pascal.Probst@med.uni-heidelberg.de (P.P.); 2Pancreas Center, The First Affiliated Hospital of Nanjing Medical University, Nanjing 210029, China; weijishu@hotmail.com (J.W.); miaoyi@njmu.edu.cn (Y.M.); 3Department of Gastrointestinal Surgery, Affiliated Hangzhou First People’s Hospital, Zhejiang University School of Medicine, Hangzhou 310006, China; 4Department of Surgery, Cantonal Hospital Thurgau, 8501 Frauenfeld, Switzerland; 5Study Center of the German Society of Surgery, University of Heidelberg, 69120 Heidelberg, Germany; dr.stevil.phd@gmail.com

**Keywords:** cancer-directed surgery, liver metastasis, overall survival, pancreatic adenocarcinoma, SEER

## Abstract

**Simple Summary:**

Pancreatic ductal adenocarcinoma is a devastating illness but guidelines consider it unresectable once metastasized. However, resection of the primary tumor is carried out in select cases and retrospective analyses indicate that this may improve survival. Even so, these analyses are limited to single centers or fail to account for biased patient selection. We overcome these limitations with a propensity score-matched SEER database analysis that reliably demonstrates surgery can extend overall survival. Furthermore, we identify prognostic factors that could aid the selection of patients for randomized controlled trials. Thus, this study paves the way for future work that aims to update treatment guidelines in accordance with surgical developments.

**Abstract:**

Guidelines do not recommend resection surgery for oligometastatic pancreatic ductal adenocarcinoma (PDAC). However, reports in small samples of selected patients suggest that surgery extends survival. Thus, this study aims to gather evidence for the benefits of cancer-directed surgery (CDS) by analyzing a national cohort and identifying prognostic factors that aid the selection of candidates for CDS or recruitment into experimental trials. Data for patients with PDAC and hepatic metastasis were extracted from the population-based Surveillance, Epidemiology, and End Results database (SEER). The bias between CDS and non-CDS groups was minimized with Propensity Score Matching (PSM), and the prognostic role of CDS was investigated by comparing Kaplan-Meier estimators and Cox proportional hazard models. A total of 12,018 patients were extracted from the database, including 259 patients who underwent CDS that were 1:1 propensity score-matched with patients who did not receive CDS. CDS appeared to significantly prolong median overall survival from 5 to 10 months. Multivariate analysis revealed chemotherapy as a protective prognostic, whilst survival was impaired by old age and tumors that were poorly differentiated (Grades III–IV). These factors can be used to select patients likely to benefit from CDS treatment, which may facilitate recruitment into randomized controlled trials.

## 1. Introduction

Pancreatic ductal adenocarcinoma (PDAC) is the most common malignant tumor of the pancreas, accounting for 90% of all pancreatic cancers [1]. In the United States, it is estimated that PDAC is the 10th most commonly diagnosed cancer (57,600 new cases in 2020), but the 4th leading cause of cancer death (47,050 cancer deaths in 2020) [2]. Furthermore, PDAC is expected to be the second leading cause of cancer death by 2030 [3]. On a worldwide scale, it is the 12th most commonly diagnosed cancer, and the 7th leading cause of cancer death in 2020 [4,5].

The only curative chance for PDAC patients is surgical resection followed by chemotherapy, yet conventional guidelines regard oligometastatic PDAC as a contraindication for surgery [6,7]. As such, only 10–20% of PDAC cases are diagnosed early enough for upfront surgical resection [8] since a lack of early symptoms coupled with aggressive cancer biology lead to most cases being metastatic at diagnosis [9]. However, there have been tremendous advances in pancreatic surgery, systemic therapy, and chemotherapy over recent decades. For instance, 5-year survival for patients receiving curative treatment has increased by an order of magnitude since the early 2000s [10,11]. Furthermore, there have been drastic survival improvements in resected metastatic PDAC patients that receive chemotherapy [12,13]. Given such advances, cancer-directed surgery (CDS) for oligometastatic PDAC is sometimes performed incidentally or in highly selected patients [14].

Despite its deviation from guidelines, surgical resection is associated with improved survival in oligometastatic PDAC patients [15,16,17]. For example, a retrospective cohort study with 85 patients revealed that CDS had low risks of surgery-related morbidity/mortality and was associated with improved long-term survival relative to palliative treatment [18]. Nevertheless, these single-center studies called for larger sample sizes drawn from multiple centers. To this end, recent studies have used the Surveillance, Epidemiology, and End Results database (SEER) to increase sample size and found that CDS of the primary tumor in metastatic PDAC patients was associated with an improvement to overall survival [19,20]. However, because CDS of the primary tumor is not standard treatment, the decision to perform surgery on oligometastatic PDAC patients may be the result of careful patient selection by surgeons. This non-random bias may influence comparisons of overall survival between patients treated with and without CDS, and this bias was unaccounted for in early analyses of the SEER database. Furthermore, these studies considered PDAC patients across multiple sites of metastasis, which may have introduced further confounds into the analysis. Consequently, surgeons still lack evidence-based selection criteria to identify the most appropriate candidates for resection surgery. This is a critical issue for two reasons. First, clinicians could propose CDS to more patients if they had evidence to expect a benefit over palliative treatment. Secondly, updating the guidelines in accordance with modern developments requires reliable evidence from randomized controlled trials (RCT), which in turn requires justified selection of patients in spite of the guidelines. For example, HOLIPANC [21] and METAPANC [22] are two proposed RCTs that have yet to recruit any patients due to the difficulties of justifying CDS over palliative chemotherapy.

Thus, the present study aims to improve SEER-based validations of survival in oligometastatic PDAC patients that receive CDS by (i) controlling potential confounds using propensity score-matching (PSM) and (ii) limiting subjects to those with metastasis in the liver. The focus is on liver metastases because it occurs in a large proportion of patients [23]. With these improvements, we hope to provide reliable survival estimates for CDS alongside other prognostic indicators. These results could be used to generate testable hypotheses for rigorous clinical experiments that aim to identify the PDAC patients most likely to benefit from CDS.

## 2. Materials and Methods

### 2.1. Data Source

The SEER program was established in 1973 and is supported by the National Cancer Institute (NCI) of the USA [24]. Nowadays, the SEER Program captures reported cancer cases from 19 U.S. geographic areas, representing 34.6% of the population (https://seer.cancer.gov, accessed on 30 January 2021). The data used in this study are publicly available and exclude identifying information on individual patients. Therefore, there is no requirement for written informed consent from patients or approval from an Institutional Review Board. This study collected data with the program SEER*Stat (Version 8.3.5. seer.cancer.gov/seerstat: Surveillance Research Program, National Cancer Institute, 2018), using research data submitted to SEER by November 2017 [25]. Because the SEER database is a large, population-based cancer registry with patient-level data, results can be better extrapolated to the general population than studies made in single centers.

### 2.2. Study Cohort

Due to database limitations, our analysis was restricted to a sample from 1 January 2010–31 December 2015. The SEER*Stat software was used to identify PDAC patients with liver metastasis and applicability for CDS (Figure 1). First, cancer patients with a primary site in the pancreas were retrieved using the topographical codes from the International Classification of Diseases for Oncology (ICD-O-3: C25.0–C25.3, C25.7–C25.9) [26]. Here, patients with tumors that originate from the pancreatic Islets of Langerhans (C25.4) were excluded. Second, to focus on PDAC, only patients diagnosed with ICD-O-3 histology/behavior codes of 8140/3 (adenocarcinoma) or 8500/3 (infiltrating duct adenocarcinoma) were included [26,27,28]. All patients without microscopic confirmation of PDAC diagnosis were excluded. Third, the focus was brought to liver metastases at the time-point of diagnosis by excluding patients with metastases to other distant organs such as the lung, brain, and bone. Only data from the time period after 2010 were retrieved because the SEER database did not provide information on the site of distant metastasis until then. Lastly, cohort selection excluded patients with unknown surgical status and those who did not receive surgery for reasons other than PDAC. Specifically, we excluded all patients that did not have Code 0 (surgery performed) or Code 1 (surgery not recommended) according to the SEER database. This exclusion applies to patients with the following codes: surgery was contraindicated due to other conditions (Code 2), the patient died before recommended surgery (Code 5), surgery was avoided for unknown reasons (Code 6), patient or guardian refused surgery (Code 7), surgery recommended but unknown if done (Code 8), unknown if surgery performed (Code 9). These criteria establish our operational definition of oligometastatic PDAC patients as those with resectable metastasis confined to the liver (and only the liver). This is in line with currently accepted definitions of oligometastatic PDAC as the presence of one metastatic site (liver or lung) with ≤4 metastases and treatable with minor resection/ablation accompanied by resection of the primary tumor [29,30]. The SEER database does not provide information on the number of metastases, but we assume that patients selected for surgery likely had 1–4 macroscopic metastases on the basis of single-center retrospective studies that included this information [18]. Ultimately, the effect of CDS on the primary PDAC lesion in the setting of liver metastasis was evaluated by categorizing patients into (a) those who received CDS of cancer (CDS Group), and (b) those who did not (No-CDS Group). The SEER classification used to define surgery was ‘resection of all macroscopically evident sites of cancer for curative purpose’ [31]. We thus assume that resection was performed on both the primary tumor and all macroscopic metastases.

### 2.3. Variables Collected

The following parameters were collected from the sample: (i) age at diagnosis, (ii) sex (Female/Male), (iii) ethnicity (White/Black/Other), (iv) marital status (Single/Married/Divorced/Widowed/Other-Unknown), (v) insurance status (Any Medicaid/Insured/Non-Specific Insurance/Uninsured-Unknown), (vi) primary site of the tumor (Pancreatic Head/Body-Tail/Other), (vii) tumor differentiation grade (I-II/III-IV/Unknown), (viii) pathological primary tumor T-Stage according to AJCC 7th ed. [32] (T0-T2/T3/T4/Unknown-NA), (ix) pathological primary tumor lymph node stage according to AJCC 7th ed. [32] (N0/N1/Unknown-NA), (x) receipt of chemotherapy (Yes/No), (xi) directed surgical resection of all macroscopically evident cancer sites (Yes/No), (xii) overall survival (OS) in months (duration from diagnosis to death from any cause). We only used data up until 2015 because the SEER database did not include data for chemotherapy, tumor N-stage, or tumor T-stage after this date. This is critical because our model comparison analyses do not tolerate missing data (records from 2015–2017 available on request). The SEER database did not provide the regimen of chemotherapy nor the quality of life. The last date of follow-up was on 31 December 2015.

### 2.4. Propensity Score-Matching (PSM)

The propensity score was defined as the likelihood of undergoing CDS (on a scale from 0 to 1), given individual characteristics. It was obtained from a logistic regression model that factored the independent correlations of all retrieved variables (i–x) on CDS status (xi). Subjects were matched by propensity score to their nearest neighbor in a 1:1 ratio without replacement. This mitigated the selection bias for particular patients to receive CDS by comparing survival outcomes between matched groups of CDS and No-CDS patients [33,34]. Validation of PSM was achieved by comparing the CDS and No-CDS groups for each observed variable before and after PSM. Continuous variables were compared with unpaired student *t*-tests, and categorical variables were compared with χ^2^-tests.

### 2.5. Survival Analysis

Overall survival was estimated using the Kaplan–Meier method (with right-censored data for those who died after data submission or dropped out). The difference in median survival between surgical groups was examined using the log-rank test. Cox proportional hazards models (with ties handled by Breslow approximation) were fitted for all predictor variables (i–xi) using the forward-stepwise-selection procedure from Ekman et al. [35]. This procedure generated 12 models, from a null model with no factors to a full model with all 11 factors. Thus, we used an information-theoretic framework to find the best explanatory models from the full set [36,37]. Specifically, the corrected Akaike Information Criterion (AICc) was calculated for each model, which indexes the amount of information provided by a model whilst penalizing it for being overloaded with factors. The AICc values were used to select a 95% confidence set, which is the set of models 95% likely to contain the best-approximating model for the data from all those considered. We averaged the hazard-ratio estimates for CDS and other predictors across the 95% credible set (weighted by AICc) to infer prognostic indicators of survival. Data analysis was performed using Stata/MP for Windows (version 13.1, StataCorp LLC, College Station, TX, USA, 2013) and R (version 4.0.3, R Core Team, Vienna, Austria:, 2020) with library MuMIn (version 1.43.17, Kamil Bartoń, Krakow, Poland).

## 3. Results

### 3.1. Selection of Study Cohort and Propensity Score Matching

A total of 12,018 patients with microscopically confirmed PDAC and hepatic metastasis at the time of diagnosis were extracted from the database (Figure 1). The median age at diagnosis for the total study population was 67 years (interquartile range 59–75 years), with a male to female ratio of 1.2:1. Of these patients, 259 (2.2%) underwent CDS of the primary tumor, while the remaining 11,759 (97.8%) did not. Patients in the CDS group had an average propensity score of 0.02 ± 0.04, whilst those in the No-CDS group had an average propensity score of 0.14 ± 0.10. After 1:1 matching patients from the No-CDS group with the CDS group, there were 259 patients in each group and both groups had an average propensity score of 0.14 ± 0.10 (Figure 2).

Table 1 shows the demographic and clinical characteristics of patients in the full and reduced datasets. The comparisons without PSM show that baseline characteristics were significantly unbalanced between the two groups for multiple covariates. However, covariates did not significantly differ between groups after PSM. Thus, PSM appears to minimize potential confounds.

### 3.2. Survival Outcomes after Propensity Score-Matching

The prognostic impact of CDS was investigated by comparing median overall survival between PS-Matched groups (*n* = 259 per group). The median overall survival for patients receiving CDS (8–12 months) was significantly greater (log-rank test: χ^2^_1_ = 34.0, *p* < 0.001) than patients in the No-CDS group (4–7 months; Figure 3 and Figure 4a). Furthermore, estimated 1-year survival rates were around 90% greater for CDS patients (CDS [95CI]: 0.40 [0.34–0.47]; No-CDS: 0.21 [0.16–0.27]), whilst 3-year survival was improved by an order of magnitude (CDS: 0.13 [0.08–0.19]; No-CDS: 0.01 [<0.01–0.07]; Figure 4a).

An IT-AIC approach was used to estimate the effect of CDS in a multivariate setting and identify additional prognostic factors that could aid in selecting patients for CDS [18]. According to AICc, there was no single definitive model that could best explain overall survival (Table 2). The top-ranked model included five factors and was 42% likely to be the best-approximating model of those considered. To improve expected predictive accuracy whilst keeping overfitting low, we considered a ‘confidence set’ of four models that together were 96% likely to contain the best model. These models imply that the following factors were informative for predicting survival: (i) chemotherapy, (ii) CDS, (iii) age, (iv) tumor grade, (v) tumor N-Stage, (vi) ethnicity, (vii) insurance status.

We derived estimates of the hazard ratios for each of these factors by averaging the estimates from each model in the confidence set, with contributions weighted by AICc (Figure 5). These estimates imply that patients receiving CDS had improved rates of survival (Figure 5), and CDS was certain to be a factor in the best model (Table 2). Of the other factors, survival was also clearly improved by chemotherapy, and negatively impacted by age (≥70 yo) and poor differentiation of the tumor (Grade III-IV) (Table 2; Figure 4b–d and Figure 5). Tumor N-Stage was also identified as an informative factor, although its influence was unclear. That is, N-Stage was 74% likely to be included in the best-approximating model, but the average hazard ratio of N1 patients did not greatly deviate from parity with N0 patients. That being said, Figure 4e suggests that differences in survival between N0 and N1 patients emerge in combination with CDS.

Of the other factors, ethnicity and insurance status were, respectively, 30% and 5% likely to be included in the best-approximating model (Table 2). Furthermore, the confidence intervals for their hazard ratios did not deviate from equivalence with reference categories (Figure 5). In any case, it appears that none of (i) primary tumor location, (ii) tumor T-Stage, (iii) marital status, or (iv) sex provided any meaningful predictive accuracy once other factors were considered (Table 2).

## 4. Discussion

A comparison of matched cohorts from the SEER database indicated that PDAC patients with liver metastasis who received CDS had significantly longer overall survival than patients without CDS. The analysis also revealed chemotherapy, age, and tumor differentiation as meaningful prognostic indicators. These results imply that CDS is most effective when patients are younger than 70, receive chemotherapy, and have well-differentiated tumors. However, CDS may also provide meaningful improvements to survival for patients with otherwise poor prognostic factors, and is thus worthy of consideration in a range of circumstances.

### 4.1. Comparison to Previous SEER Analyses

These results align with previous analyses of the SEER database, which also demonstrate the benefits of surgical resection for metastatic PDAC [19,20]. However, these other reports identify a suite of partially unspecific and heterogeneous patient-related variables associated with survival that were not identified in our primary analysis. These prognostic factors included sex, marital status, and site of primary cancer, which were all revealed as uninformative in our multi-model information-theoretic framework. The divergence in results may partly be attributed to the employment of PSM, for which our comparisons of full and reduced datasets identified many factors as potential confounds. In other words, there is possibly a bias in the selection of particular patients to receive CDS and our analyses minimized this bias. Our studies also differed in the respect that we focused specifically on patients with only liver metastasis at diagnosis, whereas the other studies had broader samples of pancreatic cancers and metastatic PDAC. Thus, some differences may be attributable to the sample under analysis.

### 4.2. Limitations of SEER Analyses

The observational analysis of the SEER database has the inherent limitation of low data resolution for clinically significant variables that could be critical to clinicians screening patients for CDS and/or overall survival. These factors include details on chemotherapy regimens, the extent of metastases, and tumor markers such as CA 19-9, alongside characteristics such as performance status, preoperative comorbidities, or postoperative complications. In other words, our SEER-based analysis improves over previous case-series reports with a larger sample, but at the cost of less information on clinical details. Hence, it is important to consider the current broad-scale results in combination with the finer-scale results of previous analyses in order to properly identify important prognostic factors for prospective RCTs.

The limitations of the SEER database constrain estimations for the prognostic role of chemotherapy, as the timing and nature of the therapy were not included in the classification. However, our sample covered a window from 2010 to 2015, thereby allowing some assumptions based on common treatments at the time. In this respect, modern combination therapies like FOLFIRINOX or Gem/nab-paclitaxel are implausible. Our previous case-series report of 85 PDAC patients with hepatic metastases at the University of Heidelberg between October 2001 and May 2014 showed that surgical patients were taking adjuvant treatments, with most patients using gemcitabine [18]. Regardless of the specific regimen, our finding that outdated chemotherapy treatments can improve survival outcomes for surgical patients implies that we can expect greater effects from modern treatments. Even so, analyzing the relation between chemotherapy regimen and survival outcome in metastatic PDAC patients receiving surgery remains an important target for prospective studies.

Relatedly, the data shed no light on the role of neoadjuvant chemo(radio)therapy for CDS in oligometastatic PDAC patients. Understanding the role of neoadjuvant therapy is important because its use is increasing, but there is no clear evidence for its effects in supporting resection surgery in oligometastatic PDAC. On the one hand, a pilot randomized controlled trial of neoadjuvant therapy in resectable and borderline-resectable PDAC cases showed improved R0-resection rates but no improvements to overall survival [38]. Furthermore, initial chemotherapy may preclude detection of occult hepatic metastases that are only evident during surgical exploration. On the other hand, we have observed improved survival with secondary PDAC-resections that follow neoadjuvant treatment for patients with initially ‘unresectable’ tumors [39]. In any case, initial chemotherapy is unlikely to make resection unnecessary when optimizing tumor control and prognosis [40]. Even so, future analyses of patient databases should incorporate initial chemo(radio)therapies when the data is available to better supplement the forthcoming research on this therapeutic approach for PDAC.

The SEER database further lacks information on the extent of liver metastasis. However, we can assume that the patients included in our study had small numbers of localized macroscopic metastasis sites in the liver. First, our study design specifies that metastases for all patients were limited to the liver and no other organ. Second, we can assume that patients receiving surgery had macroscopic cancer sites owing to the classification of surgery in the SEER database as “resection of all macroscopically evident sites of cancer” [31]. Third, we can assume that these sites were localized and not vast or diffuse because the latter circumstances are not conducive to resection surgery with a curative intention. These assumptions are supported by our previous case-series report, in which 86% of cases were atypical resections of 1–4 subcapsular lesions [18]. The average diameter of the largest resected metastasis across patients was <1 cm in 43% of patients and 1–2 cm in 31.7% of patients. Only three patients in the cohort displayed >3 metastases. According to this case-series study, neither the number nor size of liver metastases showed a significant influence on survival. In any case, prospective designs for RCTs are anyway likely to exclude patients with too many metastatic sites, e.g., [41].

Lastly, our SEER analysis did not include CA 19-9 measurements. In this respect, the 85 patients in our case-series study displayed a median CA 19-9 level of 165 U/mL, with 19% of patients exceeding a 1000 U/mL threshold [18]. Furthermore, the analysis indicated that pre-operative CA 19-9 levels did not significantly influence survival outcomes. In any case, it is likely that CA 19-9 measurements will still play a role in future clinical trials concerning CDS in PDAC treatment. For instance, decreases in CA 19-9 levels are specified as an indicator for a response to induction-chemotherapy in the design of an upcoming RCT [41]. Furthermore, CA 19-9 is included in definitions of oligometastatic PDAC [30] and used as a prognostic factor to stratify tumors by resectability and monitor for recurrence [42,43,44]. However, upcoming RCTs may need to consider that the use of CA 19-9 markers is limited by a lack of expert consensus regarding threshold values, especially in cases of combined chemotherapy and resection surgery [30,44,45]. Additionally, 5 to 10% of patients can be expected to be Lewis-negative and display little to no CA 19-9 secretions, and must thus be screened with alternative markers [46]. Thus, it is unfortunate that we cannot provide further insight into the prognostic role of CA 19-9 levels, but the existing knowledge gap suggests that it should be tracked as part of forthcoming RCTs.

Overall, some limitations of the SEER database can be compensated for by considering analytical approaches from a finer scale (e.g., single-center retrospective studies). However, retrospective study designs of all scales still carry potential selection biases alongside a lack of full control, and these limitations can only be addressed with an RCT approach.

### 4.3. The Need for Evidence-Backed Guidance When Selecting Patients for CDS

The majority of PDAC patients treated with resection surgery are already micro-metastatic and likely to experience recurring metastases [47]. Thus, eventual treatment outcomes may depend on the control or eradication of micro-metastases and not only the removal of the gross disease. Currently, surgeons are highly biased when selecting metastatic patients for resection because it subverts guidelines. Consequently, particular patients are more likely to receive surgery than others [19,48]. The current need is to provide evidence-backed guidance when selecting metastatic PDAC patients based on their tumor biology and likelihood to benefit from a therapeutic course involving CDS.

We provide reliable estimates for survival in PDAC cases with hepatic metastasis, improving on previous attempts [15,16,17,18,19,20] by excluding other sites of metastasis and reducing sources of bias with PSM and model averaging. Our results imply there is less apparent need to consider factors such as sex, tumor T-Stage, insurance status, marital status, ethnicity, or primary tumor location. Instead, our results suggest that survival odds are best for patients that (i) undergo chemotherapy, (ii) are below 70, and (iii) have well-differentiated tumors (Grades I-II).

These estimates provide the first elements toward building evidence-backed patient selection criteria for CDS. This is critical for the design of randomized controlled trials that aim to thoroughly determine the clinical effects of CDS. Such studies require justification for patient inclusion/exclusion, alongside guidance in generating hypotheses and defining the scope and boundaries for the study. Furthermore, promising estimates for improved survival can help convince prospective patients into joining a trial that involves non-standard treatments. For instance, the HOLIPANC [21] and METAPANC [22] studies aim to provide CDS for patients that have limited metastases and a positive response to initial chemotherapy. Our results justify the central importance of chemotherapy, but also suggest that experimental design could consider patient age as well as the differentiation of the primary tumor and its stratification into lymph nodes. Thus, the results of this report can find their primary application in guiding the future studies necessary to update treatment practice surrounding metastatic PDAC.

## 5. Conclusions

Forty years ago, the role of surgery in treating disseminated colorectal cancers (CRC) was confined to “some form of palliation” [49]. Nowadays, resection surgery is a standard of care; a quarter of patients with CRC present with liver metastases during the disease [2,50,51,52], and surgical resection is regarded as the optimal curative therapy [53]. Similarly, CDS is employed despite hepatic metastasis when treating some types of pancreatic neuroendocrine tumors (PNETs) [54,55,56]. Up to 30% of PNET patients have liver metastasis at diagnosis [57], and resection with curative intention is recommended for G1/G2 PNETs with Type I/II liver metastasis [58,59].

Despite the recognized benefit of CDS in such malignancies, many studies demonstrate that resection is underutilized when treating PDAC. Even in patients with early-stage PDAC, the resection rates are reported to range from 28% to 52% [60,61]. For patients with local stage III PDAC, the resection rate is only around 10% [62]. At the extreme, Stage IV PDAC cases with distant metastasis to the liver are regarded by the guidelines as unresectable.

The evolution of resection treatment beyond the primary tumor stage involves advancements in both surgery and chemotherapy. There have been tremendous advances in PDAC surgical techniques in recent years, and now resection of hepatic metastases is considered safe in some specialized institutions [63]. Further, the benefits of surgery are best observed in combination with modern adjuvant chemotherapy regimens e.g., [11,64]. Other developments include locally-ablative treatments like microwave ablation (MWA) and radiofrequency ablation (RFA), which have been established for the treatment of liver metastasis in a variety of tumors. But these treatments remain to be sufficiently proven in a metastatic PDAC setting and are thus considered complementary to surgery and systemic therapy [65]. The caveat to an extended treatment approach is increased physical stress to the patient. We could only indirectly consider the physical condition in terms of age, and thus propose future studies to include specific measures of performance [30].

Our analysis of the SEER database suggests that CDS can prolong the survival of oligometastatic PDAC patients (i.e., patients with metastasis only to the liver) and should be considered as a treatment option in selected patients. However, it is important to bear in mind that patient characteristics, cancer biology, and chemotherapy can also influence overall survival. Hence CDS in oligometastatic PDAC is only reasonable if embedded in multimodal, interdisciplinary diagnosis and treatment. This is because PDAC carries frequent occult metastases, and must therefore be treated both macroscopically and microscopically. This emphasizes our conclusion that surgical therapy must not impair reception of chemotherapy and needs to be part of a broader treatment scheme. In this respect, the current guidelines are outdated and there is a call for multi-center randomized controlled trials that validate new standards for practice. Our analysis provides this research agenda with appropriate targets and hypotheses.

## Figures and Tables

**Figure 1 cancers-14-00057-f001:**
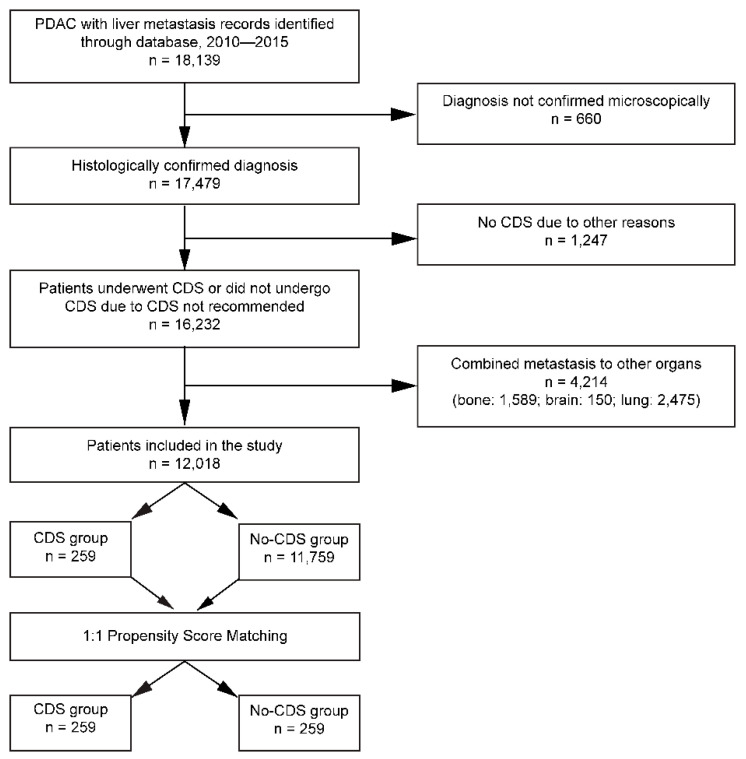
Flow chart depicting the patient selection process. PDAC: pancreatic ductal adenocarcinoma, CDS: cancer-directed surgery.

**Figure 2 cancers-14-00057-f002:**
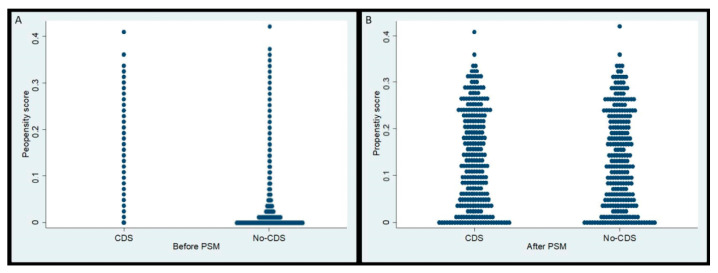
Propensity score distribution between CDS and No-CDS groups (**A**) before and (**B**) after propensity score matching. CDS: cancer-directed surgery, PSM: propensity score-matching.

**Figure 3 cancers-14-00057-f003:**
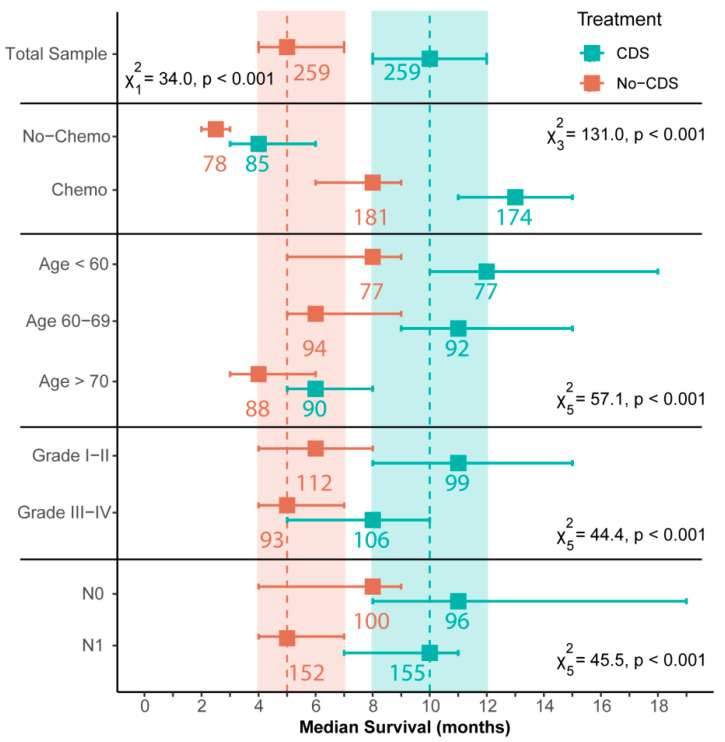
Median overall survival (months) ± 95% confidence intervals for patients in CDS and No-CDS groups. Numbers provide sample sizes. Dashed lines and shaded areas cover the median and confidence intervals for the total sample. Statistical results are for grouping-specific log-rank tests. Results are given for the total sample and by meaningful prognostic factors (see Table 2, Figure 5). Some factor levels have been removed for ease of reading. CDS: cancer-directed surgery, Grade: tumor differentiation grade, N: tumor N-stage.

**Figure 4 cancers-14-00057-f004:**
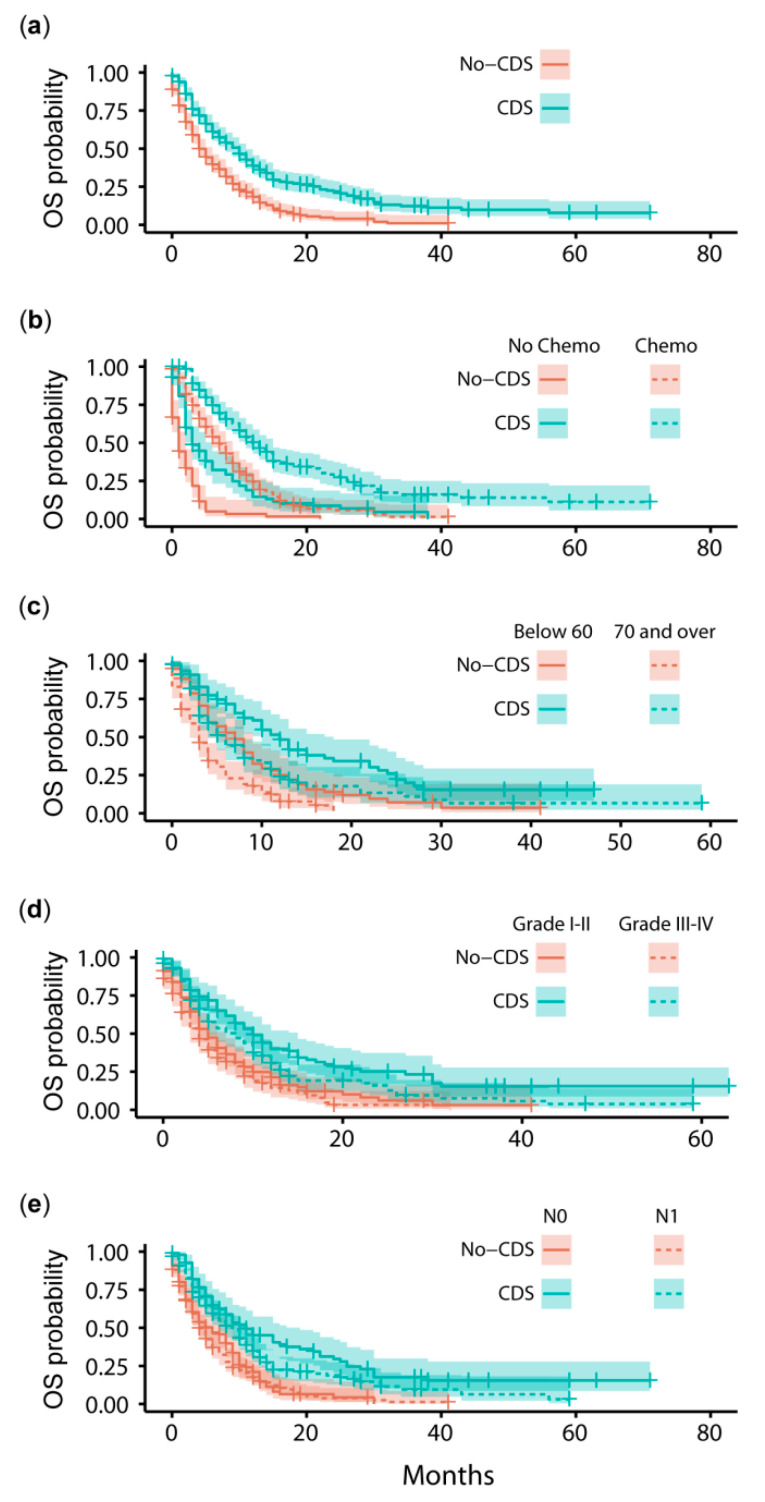
Kaplan-Meier overall survival (OS) estimates and 95% confidence intervals for patients in CDS and No-CDS groups: (**a**) in the total sample, grouped by (**b**) receipt of chemotherapy, (**c**) age, (**d**) PDAC differentiation grade, (**e**) tumor N-Stage. CDS: cancer-directed surgery.

**Figure 5 cancers-14-00057-f005:**
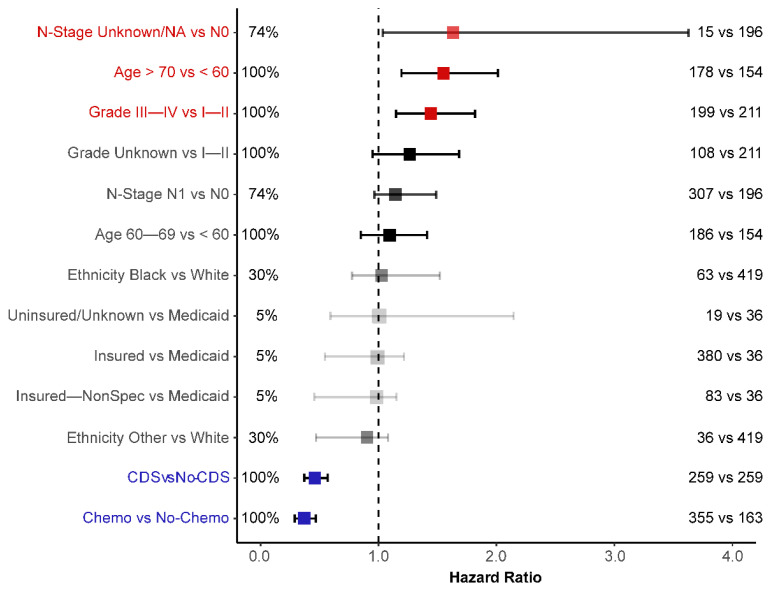
Full-model averaged Cox proportional hazard ratios with 95% confidence intervals. There is a dashed line for an equivalent hazard ratio (HR = 1). Factors are highlighted if the confidence interval for their estimate is distinct from equivalence. The percentages and opacity of estimates provide the summed AICc weight of models containing a given factor, which represents the likelihood of a factor being in the best-approximating model (Table 2). Number comparisons on the right represent sample sizes.

**Table 1 cancers-14-00057-t001:** Baseline characteristics before and after propensity score matching, showing statistical comparisons between CDS (central comparison group, highlighted in gray) and No-CDS groups (*t*-test or χ^2^ test).

Factor	Pre-PSM		CDS (*n* = 259)	Post-PSM	
	No-CDS	Comparison		Comparison	No-CDS
	(*n* = 11,759)				(*n* = 259)
Age (mean ± SD)	67.4 ± 11.2	t_12,016_ = 3.71, *p* < 0.001	64.8 ± 10.5	t_516_ = 0.23, *p* = 0.815	64.5 ± 10.9
Sex (*n*, %)		χ^2^_1_ = 0.38, *p* = 0.539		χ^2^_1_ = 0.63, *p* = 0.427	
Female	5313 (45.2)		122 (47.1)		113 (43.6)
Male	6446 (54.8)		137 (52.9)		146 (56.4)
Ethnicity (*n*, %)		χ^2^_3_ = 1.73, *p* = 0.631		χ^2^_3_ = 0.59, *p* = 0.745	
White	9336 (79.4)		209 (80.7)		210 (81.1)
Black	1591(13.5)		30 (11.6)		33 (12.7)
Other	802 (6.8)		20 (7.7)		16 (6.2)
Unknown	30 (0.3)		0 (0)		0 (0)
Marital status (*n*, %)		χ^2^_4_ = 11.88, *p* = 0.018		χ^2^_4_ = 1.20, *p* = 0.879	
Single	1689 (14.4)		31 (12.0)		34 (13.1)
Married	6570 (55.9)		170 (65.6)		164 (63.3)
Divorced	1227 (10.4)		22 (8.5)		25 (9.7)
Widowed	1628 (13.8)		30 (11.6)		27 (10.4)
Others/unknown	645 (5.3)		6 (2.3)		9 (3.5)
Insurance (*n*, %)		χ^2^_3_ = 6.16, *p* = 0.104		χ^2^_3_ = 0.52, *p* = 0.915	
Any Medicaid	1448 (12.3)		20 (7.7)		16 (6.2)
Insured	7895 (67.1)		189 (73.0)		191 (73.7)
Insured/no specifics	1900 (16.2)		41 (15.8)		42 (16.2)
Uninsured/unknown	516 (4.4)		9 (3.5)		10 (3.9)
Tumor location (*n*, %)		χ^2^_2_ = 68.43, *p* < 0.001		χ^2^_2_ = 1.41, *p* = 0.493	
Pancreatic head	4445 (37.8)		159 (63.4)		172 (66.4)
Pancreatic body/tail	4203 (35.7)		76 (29.3)		66 (25.5)
Pancreas other	3111 (26.5)		24 (9.3)		21 (8.1)
Grade (*n*, %)		χ^2^_3_ = 564.93, *p* < 0.001		χ^2^_3_ = 1.88, *p* = 0.757	
I	122 (1.0)		11 (4.3)		13 (5.0)
II	861 (7.3)		88 (34.0)		99 (38.2)
III	1243 (10.6)		104 (40.2)		92 (35.5)
IV	43 (0.4)		2 (0.8)		1 (0.4)
Unknown	9490 (80.7)		54 (20.9)		54 (19.3)
T stage (*n*, %)		Χ^2^_5_ = 218.81, *p* < 0.001		χ^2^_5_ = 2.98, *p* = 0.703	
T0	121 (1.0)		0 (0)		2 (0.8)
T1	304 (2.6)		9 (3.5)		10 (3.9)
T2	3339 (28.4)		38 (14.7)		41 (15.8)
T3	3100 (26.4)		173 (66.8)		166 (64.1)
T4	1973 (16.8)		24 (9.3)		21 (8.1)
Tx/NA ^1^	2921 (24.8)		15 (5.8)		19 (7.3)
N stage (*n*, %)		χ^2^_2_ = 123.10, *p* < 0.001		χ^2^_2_ = 0.18, *p* = 0.915	
N0	6400 (54.4)		96 (37.1)		100 (38.6)
N1	3417 (29.1)		155 (59.8)		152 (58.7)
Nx/NA ^1^	1942 (16.5)		8 (3.1)		7 (2.7)
Chemotherapy (*n*, %)		χ^2^_1_ = 12.93, *p* < 0.001		χ^2^_1_ = 0.44, *p* = 0.508	
No/unknown	5177 (44.0)		85 (32.8)		78 (30.1)
Yes	6582 (56.0)		174 (67.2)		181 (69.9)

^1^ One patient was NA. The shaded column is a means to the CDS group as the central point of comparison. On the left is a comparison with pre-PSM non-CDS patients, and on the right is a comparison with post-PSM non-CDS patients. It is there to separate the two 1v1 comparisons, rather than to compare among 3 groups.

**Table 2 cancers-14-00057-t002:** Set of models created with forward-stepwise selection, ranked by corrected AIC.

chm	cds	age	grd	n	eth	ins	loc	t	mrg	sex	K	LL	AICc	ΔAIC	AICcW	∑Wt
											8	−1954.45	3925.29	0.00	0.42	0.42
											6	−1957.04	3926.30	1.01	0.25	0.67
											10	−1952.88	3926.36	1.07	0.24	0.91
											13	−1951.29	3929.59	4.30	0.05	0.96
											15	−1949.86	3931.06	5.77	0.02	0.98
											4	−1962.01	3932.13	6.84	0.01	1.00
											18	−1948.76	3935.43	10.14	0.00	1.00
											2	−1967.80	3939.64	14.35	0.00	1.00
											22	−1947.78	3942.44	17.14	0.00	1.00
											23	−1947.73	3944.60	19.31	0.00	1.00
											1	−1990.57	3983.14	57.85	0.00	1.00
											0	−2015.38	4030.76	105.47	0.00	1.00

Shaded boxes signify the factors included within the model. Models with darker shading represent the confidence set, which is > 95% likely to contain the factors of the best-approximating model (based on ∑Wt). K is the number of parameters. LL is the log-likelihood. ΔAICc is the difference in corrected AIC compared to the top-ranked model (values < 2 indicate informational equivalence). AICcWt represents the proportional AICc weight of the model in the total set of models (values approximate the likelihood that a given model is the best of those in the set). ∑Wt is the cumulative sum of AICc weights. chm: Chemotherapy (Yes/No); cds: Cancer-Directed Surgery to resect primary PDAC (Yes/No); age: Age by class (60/60–69/≥ 70); grd: Cancer Grade (I-II/III-IV/Unknown); n: Tumor N-Stage (N0/N1/Unknown-NA); eth: Race (White/Black/Other); ins: Insurance Status (Any Medicaid/Insured/Non-Specific Insurance/Uninsured-Unknown); loc: Primary Tumor Location (Pancreatic Head/Body-Tail/Other); t: tumor T-Stage (T0-T2/T3/T4/Unknown-NA); mrg: Marital Status (Single/Married/Divorced/Widowed/Other-Unknown); sex: (Female/Male).

## Data Availability

The data presented in this study are openly available in FigShare at https://doi.org/10.6084/m9.figshare.16713052.v1 (accessed on 16 October 2021).

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
