# Peer review of "Survival Benefit of Resection Surgery for Pancreatic Ductal Adenocarcinoma with Liver Metastases: A Propensity Score-Matched SEER Database Analysis"

_cancers, 2021, doi:10.3390/cancers14010057_

Round 1

Reviewer 1 Report

Congratulation on successful approach to recent critical issue of PDAC with oligo-metastasis.

  1. There is no comments on the status of liver metastasis (single, multiple, numbers..), modality of local treatment of the liver. Please be specific on this. Or add these issues to limitations of the present study.
  2. Not only neoadjuvant treatment issue but also preoperative CA 19-9 issue also need to be included in discussion session.
  3. Improving surgical technique of pancreatectomy/ Satety of liver resection or local treatment/ introduction of new potent chemotherapeuitc agents/ patients' general condition is improving: These factors are encouraging this issue to be investigated. Please add this comments in discussion session.

Reviewer 2 Report

The study describes SEER-based validations of survival in oligometastatic PDAC patients that receive cancer-directed surgery. By far, this is the largest study analyzing this surgical approach. The paper draws  substantial conclusions and great contribution of decision making to this cohort of patients. In addition, the results of the study show meaningful improvement to survival for patients with prognostic factors. The authors provide evidence-based guidance that can lead towards clinical practice and international guidelines changes.

Reviewer 3 Report

This article focus on the role of surgery in a palliative setting, with particular interest in the impact of surgical resection on overall survival.

The article is well written, deals with a relevant topic on pancreas cancer, metodology is clear and conclusions are consistent with data presented, despite of several limitations. In particular, Propensity Score Matching is really useful in order ti minimize bias due to retrospective analysis and author's conclusions are acceptable with the aim to propose new randomized controlled Trials, in order to validate role of surgical resection in metastatic patients, after an adequate initial chemotherapy (as clearly shown in Figure 3).

I just have minor issues to clarify:

  • Line 85: the study focus on "oligometastatic PDAC"; how is defined the oligometastatic condition? Is it considered as a single organ involvement or does it depend on number of lesions? Were all selected patients considered oligometastatic? Please clarify.
  • Was surgical resection performed only on primary tumors or did selected patients undergo any treatment on metastases? Please clarify.
  • Line 296 and 306: "neoadjuvant treatment" could sound inappropriate for metastatic patients who don't have a clear surgical program; probably, "initial chemotherapy" could be more appropriate.

Reviewer 4 Report

Pausch, et al utilized the SEER database to explore the surgical benefit in patients who was diagnosed as having pancreatic ductal adenocarcinoma with liver metastasis, by use of PSM. They revealed that the CDS (cancer-directed surgery) could provide better survival in this patient cohort, which might be recommended in patients who was younger than 70, had chemotherapy and well-differentiated tumors. Basically, this study used a big data of the SEER, and is a clinically important issue in the area of pancreatic surgery. I list up some points to be discussed as below.

  1. The definition of “oligometastasis” is unclear. Please provide the definition of this disease situation in this study.
  2. The association with comment 1, when analyzing the surgical benefit in patients with liver metastasis, the extent of liver metastasis should be a big factor for the survival outcomes. Unfortunately, the information regarding this lacks in this analysis.
  3. As commented in the discussion section, chemotherapy as the prognostic factor is extremely unclear. That is, the information regarding neoadjuvant or adjuvant, regimen used, etc lacks in this study. Therefore, it is pretty hard to accept this result from this analysis.

Round 2

Reviewer 4 Report

The manuscript has been adequately revised based on the recommendations.